# Holistic Integration of Omics Tools for Precision Nutrition in Health and Disease

**DOI:** 10.3390/nu14194074

**Published:** 2022-09-30

**Authors:** Omar Ramos-Lopez, J. Alfredo Martinez, Fermin I. Milagro

**Affiliations:** 1Medicine and Psychology School, Autonomous University of Baja California, Tijuana 22390, Mexico; 2Precision Nutrition and Cardiometabolic Health, IMDEA Food Institute, CEI UAM+CSIC, 28049 Madrid, Spain; 3Department of Nutrition, Food Sciences and Physiology, University of Navarra, 31008 Pamplona, Spain; 4Center for Nutrition Research, University of Navarra, 31008 Pamplona, Spain; 5Centro de Investigación Biomédica en Red de la Fisiopatología de la Obesidad y Nutrición (CIBERobn), Institute of Health Carlos III, 28029 Madrid, Spain; 6Navarra Institute for Health Research (IdiSNA), 31008 Pamplona, Spain

**Keywords:** precision nutrition, holistic approach, genomics, epigenomics, metagenomics, transcriptomics, proteomics, metabolomics

## Abstract

The combination of multiple omics approaches has emerged as an innovative holistic scope to provide a more comprehensive view of the molecular and physiological events underlying human diseases (including obesity, dyslipidemias, fatty liver, insulin resistance, and inflammation), as well as for elucidating unique and specific metabolic phenotypes. These omics technologies include genomics (polymorphisms and other structural genetic variants), epigenomics (DNA methylation, histone modifications, long non-coding RNA, telomere length), metagenomics (gut microbiota composition, enterotypes), transcriptomics (RNA expression patterns), proteomics (protein quantities), and metabolomics (metabolite profiles), as well as interactions with dietary/nutritional factors. Although more evidence is still necessary, it is expected that the incorporation of integrative omics could be useful not only for risk prediction and early diagnosis but also for guiding tailored dietary treatments and prognosis schemes. Some challenges include ethical and regulatory issues, the lack of robust and reproducible results due to methodological aspects, the high cost of omics methodologies, and high-dimensional data analyses and interpretation. In this review, we provide examples of system biology studies using multi-omics methodologies to unravel novel insights into the mechanisms and pathways connecting the genotype to clinically relevant traits and therapy outcomes for precision nutrition applications in health and disease.

## 1. Introduction

Precision nutrition integrates information at scale by taking into account endogenous individuals’ backgrounds, but also exogenous factors including lifestyle aspects, cultural, socioeconomic and psychosocial characteristics, and food environments [1]. Thus, precision nutrition adopts a whole and dynamic scope to develop comprehensive tailored dietary recommendations for individuals and population subgroups centered on potentiating human health and nutritional wellbeing, as well as the prevention and management of chronic diseases [2,3].

Thus, the notion of precision nutrition should contemplate in-depth metabolic phenotyping using high-throughput omics technologies such as genomics (polymorphisms and other structural genetic variants), epigenomics (DNA methylation, histone modifications, long non-coding RNA, telomere length), metagenomics (gut microbiota composition, enterotypes), transcriptomics (RNA expression patterns), proteomics (protein signatures), and metabolomics (metabolite profiles) under a holistic approach (Figure 1).

However, because each method cannot individually explain entirely metabolic fingerprints, the bioinformatic integration of multiple omics skills has emerged as an innovative scope to provide a more comprehensive view of the molecular and physiological events leading to human disease [4]. Moreover, the concurrent application of these tools is helping to elucidate unique and specific phenotypes, enabling the design and implementation of personalized medicine schemes for precision health [5].

In this review, we provide examples of system biology studies using multi-omics methodologies (comprising genomics, metagenomics, epigenomics, transcriptomics, proteomics, and metabolomics) and interactions with dietary/nutritional factors to unravel novel insights into the mechanisms and pathways connecting the genotype to clinically relevant traits and therapy outcomes for precision nutrition applications in health and disease. In this regard, complex relationships between the genetic background, inherited epigenetics, nutrient utilization, microbiome-derived metabolites, and lifestyle factors may modulate gene and protein expressions, with implications for physiological processes, inflammation and metabolic phenotypes (Figure 2).

## 2. Genomics in Combination Epigenomics, Metagenomics, Transcriptomics, Proteomics or Metabolomics Tools

The integration of genomics with other functional omics analyses may significantly improve the accuracy of host genetic data for explaining health outcomes [6]. For example, using multiple machine learning algorithms, best predictors of obesity status were identified, and included single-nucleotide polymorphisms (mapped to genes such as *STXBP6*, *BBX*, *PLXDC2*, *PCDH15*, *TPH2*, *PCDH15*, *CALN1*, *FGF14*, *LRRN1*, *ACTBP2*, *RBMXP1*, and *ZNF32*) together with differentially methylated sites (in proximity to *CPT1A*, *ABCG1*, *SLC7A11*, *RNF145*, and *SREBF1* genes) and interactions with dietary factors encompassing specific foods, micronutrients, and bioactive compounds [7]. Additionally, the consortium of single-nucleotide polymorphisms in genes related to obesity and cardiometabolic diseases, low adherence to the Mediterranean diet, and harboring specific urolithin metabotypes (as biomarkers of the gut microbiota), was able to predict obesity in childhood and adolescence [8].

Intriguingly, higher milk intake in genetically lactase non-persistent individuals was reported to be associated with increased quantities of gut *Bifidobacterium* and serum concentrations of indolepropionate, a microbial-derived tryptophan metabolite inversely associated with type 2 diabetes risk [9]. Comparably, *PNPLA3* gene variants, gut bacterial features (low abundances of *Faecalibacterium* or *Prevotella*, and high abundances of *Gemmiger* taxa), and specific dietary factors (low in fiber and vitamins as well as enriched in amino acids, uric acid and purine) were all associated with different histology features in non-alcoholic fatty liver disease [10]. Moreover, relevant interrelationships between gut Prevotellaceae and an obesity-related genetic risk score determined interindividual BMI differences in women [11]. Meanwhile, it was demonstrated that the microbiomes of subjects with low copy numbers of the *AMY1* locus had enhanced capacity to break down dietary complex carbohydrates [12].

Notably, the methylation status of the *APOA2* gene was associated with the intake of saturated fat and the *APOA2* -265T>C genotype, promoting a differential *APOA2* mRNA expression between *APOA2* genotypes and modulating tryptophan and branched-chain amino acid (BCAAs) metabolic pathways [13]. Meanwhile, significant combined and interactive effects between two dietary factors related to gut microbiota (allium vegetables and overnight meal) and polymorphisms in the miRNA binding site of *IL13* gene were detected in relation to the susceptibility to colorectal cancer risk [14]. Additionally, the *LPL* variant rs13702 induced an allele-specific regulation of the *LPL* gene, affecting blood lipid traits through the disruption of miR-410 binding sites, where interactions with dietary polyunsaturated fatty acid (PUFAs) played an important role [15].

Although caution should be exercised, an exploratory analysis suggested that the maternal *FADS2* rs174575 genotype, combined with DNA methylation status in this gene, could be related to plasma fatty acid concentrations in toddlers [16]. Similarly, a meta-analysis revealed that higher *ABCA1* promoter cg14019050 methylation correlated with lower *ABCA1* expression and was concomitantly associated with the C allele of the *ABCA1* rs2246293 variant and lower circulating eicosapentaenoic acid [17]. Genomic, epigenomic, and lipidomic analyses also showed that polymorphisms and methylation sites within the *FADS1/2* region influenced the plasma levels of arachidonic acid in response to a high-fat meal in humans [18]. Of note, genomic assays of post-prandial lipidomic features after dietary fat intake identified potential biomarkers of cardiovascular risk including two polymorphisms in the *SORBS1* gene (rs12247017 and rs12240292) affecting b-Sitosterol plasma concentrations [19].

The AA genotype of the *Cdx-2 VDR* polymorphism was associated with higher methylation of the *VDR* gene promoter and lower serum levels of 25-hydroxyvitamin D in infertile men [20]. Consistently, correlations between vitamin D intake and the expression of miRNA let-7a/b varied with *VDR BsmI* genotype in an elderly cohort [21]. Moreover, the blood levels of homocysteine were influenced by the dietary intakes of methyl group donors (methionine and 5-methyltetrahydrofolate), whose interactions with methylation-related gene polymorphisms (such as 2R3R-*TS*, C1420T-*SHMT*, A2756G-*MS*, and 844ins68-*CBS*) modified the risk of adenomatous polyps, a precursor of colorectal cancer [22]. In this regard, it was demonstrated that an miRNA binding site polymorphism (rs1062980) in the iron regulatory pathway, together with dietary iron intake, may modify the risk of lung cancer tumorigenesis [23]. Likewise, the *MTHFR* C677T polymorphism influenced genomic DNA methylation in peripheral blood mononuclear cells depending on folate status, where only subjects carrying the T/T genotype with low serum levels of folate accounted for reduced DNA methylation [24]. Accordingly, DNA methylation was directly related to folate concentrations in red blood cells in subjects carrying the T/T genotype of the *MTHFR* C677T polymorphism, but not in those with wild-type [25]. Choline intake also interacted with the *MTHFR* C677T genotype to influence changes in genomic DNA methylation and DNA damage in folate-compromised Mexican American men [26].

Remarkably, prospective analyses in different population cohorts showed that the habitual intake of food source B-vitamins may modify the effects of DNA methylation-related variants at *SREBF1* and *HIF3A* genes on long-term adiposity changes [27,28]. In fact, interactions of fat intake with genetic (rs11150675), transcriptional (ILMN_1725441), and methylation (cg26663590) variations at the *NFATC2IP* locus mediated weight loss rates in response to dietary intervention [29]. However, an integrative model using microbiota and genetic information was proposed to prescribe two hypocaloric diets with different micronutrient distribution for a successful weight loss in individuals with excess of body weight [30]. Moreover, diet induced weight loss led to polymorphism-dependent modulation of miRNAs from the miR 25/93/106 gene cluster in humans [31].

Additionally, protein quantitative trait locus analyses provided evidence for distinct genetic mechanisms regulating BMI-associated proteins during diet-induced weight loss, including those associated with leptin protein expression changes [32]. Consistently, multivariate proteomic analysis using data from two clinical cohorts with obesity identified genetically driven proteins associated with low-grade inflammation, insulin resistance and dyslipidemia, which could act as endophenotypes for metabolic diseases [33]. Furthermore, BMI was associated with widespread changes in the human plasma proteome under substantial genetic control, impacting clinically relevant pathways of adiposity such as lipid metabolism and inflammation [34].

Lastly, a high dietary intake of antioxidants (α, β-carotene and α-tocopherol) protected buccal cells from telomere length (TL) shortening, depending on the genetic background of antioxidant vitamin-related genes (*BCMO1* and *ISX*) in healthy Japanese adults [35]. In addition, a higher adherence to the Mediterranean dietary pattern prevented leukocyte TL shortening among Ala allele carriers of the *PPARγ2* (rs1801282) polymorphism in subjects with high cardiovascular risk [36]. Additionally, inverse associations between TL and plasma zinc were found, especially in children carrying the homozygous mutant genotype of the *RFC* G80A (rs1051266) gene polymorphism [37]. Furthermore, telomerase RNA component genetic variants interacted with plasma monounsaturated fatty acids (MUFAs) levels, improving inflammation status and telomere attrition related with coronary heath disease [38].

## 3. Metagenomics Integrating Epigenomics, Transcriptomics, Proteomics or Metabolomics Methodologies

Metagenomic sequencing techniques have contributed to identify a number of microbial communities in the gut under different physiological and disease conditions [39]. Additional multi-omic tools analyzing gut microbial mRNA (metatranscriptomics), proteins (metaproteomics) and metabolites (metabolomics) are complementing information about the gut microbial ecology, the biological roles of uncultured microbes, and complex interactions between host, gut microbes, and environment affecting health status [40].

For instance, a clinical trial demonstrated that the daily consumption of 12 g of a prebiotic fiber supplement for 4 weeks significantly increased the abundance of several beneficial *Bifidobacterium* species and the production of health-promoting bacteria-derived metabolites in healthy individuals, with abundance of genes associated with prebiotic utilization, acetate production, and choline to betaine oxidation [41]. Additionally, normal diets of participants supplemented with either pea- or orange-fiber-containing snacks for 10 weeks correlated with abundances of genes encoding carbohydrate-active enzymes in the fecal microbiome, whose changes in turn correlated with levels of plasma proteins involved in vascular function, fibrotic responses, immune cell signaling, and obesity-associated hormonal regulators [42]. However, a very low-calorie diet intervention (800 kcal/day) in obese postmenopausal women induced changes in individual microbial taxa correlating with variations in the plasma metabolome, fecal bile acid composition, and altered gene expression pathways in adipose tissue [43]. Improvements in metabolic alterations were linked to specific microbiota genera (relative abundances of Lachnospiraceae *NK4A136* and uncultured genera of Ruminococcaceae) and fecal metabolites (cholate and cadaverine) after following a Mediterranean dietary pattern for 2 months instead of only nut consumption [44]. Consistently, the consumption of a vegetarian diet for 4 weeks significantly improved cardiometabolic risk factors and altered the relative abundance of gut microbes (dominated by several genera of Ruminococcaceae) and plasma metabolites (including l-carnitine, acylcarnitine metabolites, and phospholipids) in patients with ischemic heart disease [45]. Additionally, differential gut microbial protein expression was detected in stool samples of individuals consuming diets varying in fiber content and glycemic index for 28 days, including those implicated in production and degradation of fatty acids [46]. Interestingly, replacing beef with a chicken-based diet for two weeks largely affected the abundance of *Bacteroides* genus, and thus probably induced downregulation of immunoglobulins in feces, especially in high- and middle-BMI Chinese volunteers [47].

At the cross-sectional level, the intakes of plant-derived nutrients or artificial sweeteners in healthy individuals were associated with relevant differences in circulating metabolites (particularly bile acids) depending on gut enterotypes [48]. Similarly, gut microbiome composition influenced the relationships between soy isoflavone intake and plasma and stool metabolites, including 2-hydroxybutyrate, glycine, and liquiritigenin, with relevance in hypertension and diabetes pathogenesis [49]. Remarkably, yoghurt consumption was associated with reduced visceral adiposity and changes in gut microbiome (transient increases of *S. thermophilus* and *B. lactis* species) and fecal metabolome (elevated concentrations of 3-hydroxyoctanoic acid) in female twins [50]. Of note, chicken eaters had more diverse gut microbiota and higher abundances of *Prevotella* 2 and 9 than pork eaters, which positively correlated with fecal levels of skatole and indole [51]. Moreover, the levels of circulatory or gut metabolites were concurrently influenced by gut microbiome alterations shaped by the quality of diet consumed [52]. In addition, dietary differences among vegans and omnivores correlated with large variations in the metabolome, including co-metabolites produced by the gut microbiota [53]. In general, higher occurrence of potentially beneficial host microbiome metabolites (i.e., short- and medium-chain fatty acids and their derivatives) have been found in vegans compared to omnivores [54]. Likewise, vegan and vegetarian diets were associated with increased abundance of microbial genes/proteins involved in cell motility, nutrient breakdown and transport, and the synthesis of essential amino acids and vitamins [55]. Specifically, vegetarians showed low levels of circulating BCAAs and upregulation of the gut microbial pathway implicated in the degradation and utilization of BCAAs [56]. Accordingly, multi-omics analyses revealed that Indian subjects presented unique gut microbiome and serum metabolome profiles compared to other populations, which were associated with specific dietary patterns [57]. Meanwhile, high-level adherence to a Mediterranean diet (based on plant foodstuffs) was associated with increased levels of fecal short-chain fatty acids and favorable microbiome-related metabolomic profiles [58].

Furthermore, it has been demonstrated that gut microbiota may mediate the effects of diet on the host health via mechanisms targeting the epigenome [59]. Thus, microbial metabolites of diet (i.e., phenolic acids, isothiocyanates, and short chain fatty acids) may influence epigenome status may altering the expression of epigenetically active enzymes including DNA methyltransferases, histone acetyltransferases, deacetylases and demethylases [60]. In this context, the impact of common dietary patterns on the gut microbiota composition and the host epigenome status has been recently reviewed [61]. However, because most evidence comes from animal studies and in vitro assays, further clinical trials are still required to analyze the nutrition-microbiota-epigenetic axis and applications in precision nutrition approaches.

Additionally, it was evidenced that imbalances in intestinal microbiota due to diet may contribute to the development of different pathologies by impairing the expression of miRNAs [62]. Accordingly, a poor diet quality was associated with a higher risk of mild cognitive impairment, which could have been mediated by microbiota composition (abundance of Proteobacteria and Gammaproteobacteria) and miRNA expression (hsa-let-7g-5p, hsa-miR-107, and hsa-miR-186-3p) in middle-aged and elderly Chinese population [63]. Of note, interactions between the abundance of bacterial species (i.e., *Bacteroides eggerthi*) and circulating miRNAs (miR-130b-3p, miR-185-5p and miR-21-5p) were found in humans in relation to obesity [64]. Moreover, an integrated analysis using 25 miRNAs, 25 taxa and 7 dietary nutrients was able to clearly discriminate vegan, vegetarian, and omnivore dietary patterns in healthy individuals [65]. Lastly, dietary plant-derived miRNAs (xenomiRs) appear to modulate gut microbiota composition, influencing gut epithelial barrier permeability and related gastrointestinal health [66].

## 4. Nutritional Relationships between the Epigenome, Transcriptome, and the Metabolome

Epigenome landscapes play an important role in determining cell phenotypes via regulation of gene expression [67]. In turn, certain nutrients may induce epigenetic modifications such as DNA methylation, probably modifying the expression of key genes associated with physiologic and pathologic processes [68]. Thus, some observational and intervention studies have analyzed interactions between nutrition, epigenome, and transcriptomic signatures on health outcomes. For example, associations between dietary folate deficiency, *CAMKK2* methylation and expression levels, and insulin resistance status were reported in subjects with obesity [69]. Additionally, gestational fish intake was related to changes in the methylation and expression levels of the *FADS1/2* and *ELOVL5* genes, with impact on allergy development in early childhood [70].

Consistently, fatty acid supplementation with 4 g/day of either n-3 PUFAs or olive oil (OO) for 8 weeks altered the methylation and the transcript levels of the *FADS2* and *ELOVL5* genes in peripheral blood mononuclear cells (PBMCs) of adults suffering renal disease [71]. Likewise, it was demonstrated that dietary supplementation of 5.7 g/day of n-3 PUFAs or 6 g/day of extra virgin olive oil (EVOO) for 4 weeks induced DNA methylation changes in leukocytes in trained male cyclists, potentially via the modulation of DNA methyltransferases (*DNMTs*) mRNA expression [72]. By contrast, daily supplementation with 200 mg of monomeric and oligomeric flavanols from grape seeds for 8 weeks modulated the expression of genes associated with cardiovascular disease pathways, without parallel changes in DNA methylation states [73].

Interestingly, an integrated transcriptomic and epigenomic analysis identified differential DNA methylation and expression levels of the *CD44* gene on PBMCs depending on the success to the RESMENA (moderately high-protein content) weight-loss program [74]. In addition, the beneficial effects of consuming a very low-calorie ketogenic diet (600–800 kcal/day) for six months on obesity measurements involved methylome and transcriptome changes in *ZNF331* and *FGFRL1* genes in blood leukocytes [75].

Furthermore, epigenomics act as a mechanistic link between energy metabolism and control of gene expression (known as metaboloepigenetics), where a number of dietary metabolites (including SAM, acetyl-CoA, NAD+, and ATP) modulate the activity of epigenetic enzymes regulating transcriptional rates as necessary to maintain cell homeostasis [76]. Therefore, epigenetic marks (such as DNA methylation, posttranslational histone modifications, and nucleosome position) have the capacity to integrate the expression state of chromatin with the metabolic state of the cell [77]. In this regard, maternal nutrient availability of diet-derived methyl donors (folate, coline, betaine, and methionine) and cofactors (vitamins B2, B6, and B12) is critical for DNA methylation reactions through 1-carbon metabolism, which impacts gene expression and health outcomes in offspring [78]. Moreover, the risk for non-communicable diseases (NCDs) in adulthood can be programmed by early nutrition through alterations in methylation/expression patterns of key genes implicated in various metabolic pathways during development, and persisting into adulthood [79]. Hence, the importance of applying personalized nutrition approaches targeting the epigenome, transcriptome, and metabolome to prevent and treat NCDs since early stages of fetal life has been highlighted [80].

## 5. Metabolomics, Proteomics, and Transcriptomics Interplays

The analysis of proteins and metabolites associated with nutritional features may provide insights into the molecular mechanisms mediating diet-related disease, with preventive and management applications [81]. For instance, three dietary patterns, the Mediterranean-style diet, the Dietary Approaches to Stop Hypertension diet, and the Alternative Healthy Eating Index were unique metabolome and proteome signatures involved in important physiological pathways such as cellular metabolism and immune response within the Framingham cohort [81]. Additionally, plasma metabolomics and proteomics analyses revealed subtle multiple processes related to metabolism, oxidation and inflammation after a postprandial dietary challenge as demonstrated by changes in the concentrations of metabolites, proteins and clinical chemistry parameters in overweight subjects [82]. Interestingly, it was demonstrated that a nutrigenomic intervention with a nutritional supplement containing selected bioactive compounds (including polyphenols, alpha-tocopherol, vitamin C, and n-3 PUFAs) for five weeks affected inflammatory processes, oxidative stress, and metabolism in healthy overweight men, as evidenced by an integrated analysis of plasma metabolites/proteins, and gene expression profiles [83].

## 6. Conclusions

The combination of diverse types of biological data from genomics, epigenomics, metagenomics, transcriptomics, proteomics and metabolomics is expanding our current understanding of the complexity and diversity of human metabolism, as well as yielding profound insights into disease pathogenesis. Hence, this knowledge is allowing for the identification and characterization of potential molecular targets and active biomarkers involved in many nutritional disorders, including obesity, dyslipidemias, fatty liver, insulin resistance, and inflammation. Additionally, it is expected that the application of integrative omics approaches could be useful not only for stratifying patients for risk prediction and early diagnosis purposes but also for guiding precision disease treatments and prognosis under a holistic scope. For instance, the prediction of obesity risk and weight loss has considerably improved when using genomics, epigenomics, metagenomics and transcriptomics signatures instead of single omics approaches. Additionally, integrative genomic, epigenomic, and metabolomic analyses have better characterized post-prandial lipidomic features as potential biomarkers of nutrient intakes (including polyunsaturated fatty acids and B-complex vitamins) and subsequent cardiovascular and cancer risks. Moreover, the combined application of metagenomics, epigenomics, metabolomics, and proteomics methodologies has allowed us to discriminate different dietary patterns including vegan, vegetarian, and omnivore regimes and their implications for health status. In addition, the early programmation of chronic diseases in adulthood can be tackled by precision nutrition strategies targeting the epigenome, transcriptome, and the metabolome in preconception and pregnancy stages. However, despite these scientific advances, more evidence in these research areas is still necessary before precision nutrition can be implemented in clinical practice and public health settings around the world. Furthermore, some challenges include ethical and regulatory issues, the lack of robust and reproducible results due to methodological aspects (type of samples analyzed, standardization of procedures, population characteristics), the high cost of omics methodologies, and high-dimensional data analyses and interpretation.

## Figures and Tables

**Figure 1 nutrients-14-04074-f001:**
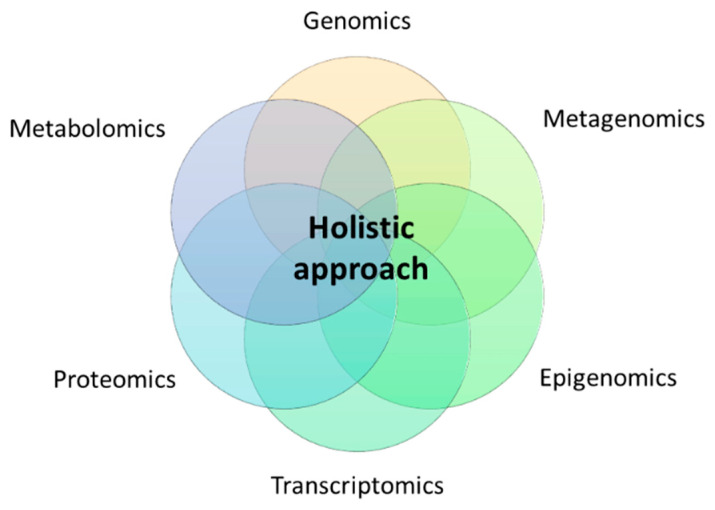
Holistic approach for precision nutrition using multi-omics technologies.

**Figure 2 nutrients-14-04074-f002:**
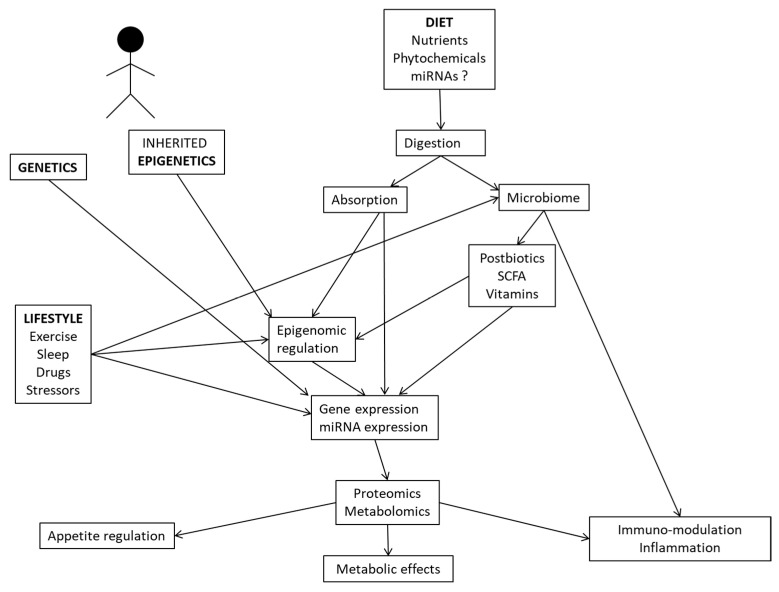
Interrelationships between multi-omics tools and environmental factors influencing human metabolism and physiological functions.

## Data Availability

Not applicable.

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
