# Peer review of "Holistic Integration of Omics Tools for Precision Nutrition in Health and Disease"

_nutrients, 2022, doi:10.3390/nu14194074_

Round 1
Reviewer 1 Report
In the submitted manuscript, the authors try to show the importance of omics tools for precise nutrition. They describe papers on a wide range of diseases and patient groups, jumping from the influence of genotype or gene expression to the involvement of miRNAs or methylation and nutrient levels. In my opinion, the article would be of greater value to the reader if the authors focused, for example, on one disease or metabolic pathway or nutrient, and cited publications from such a narrow area. Reading the paper, I had the impression of chaos. There is no doubt that the topic is important, but it is also difficult to draw conclusions covering the entirety of these various data, especially as the authors themselves indicate the methodological discrepancies in the cited works.
Author Response
Response: We understand the reviewer's point of view. For this reason, we have divided the review in different chapters according to the type of data that are combined in each study. However, this review has been designed to show examples of how different omics tools can be combined to provide a more comprehensive scope of the biological processes involved in human health status and disease development. Given the heterogeneity of the results and the number of available studies, it is not feasible to focus on a specific disease or metabolic pathway. Instead, we try to point out how a holistic approach incorporating a combination of omics information can be more informative to better characterize patients and prescribe more personalized treatments. In any case, we now incorporate the term “in health and disease” in the title to cover the areas of study mentioned throughout the text.
Moreover, in order to address the recommendation of the reviewer, we have selected some examples focused on specific topics and highlighting how they can be addressed by the application of multi-omics technologies for precision nutrition, which were now mentioned in the conclusions: “For instance, the prediction of obesity risk and weight loss have considerably improved when using genomics, epigenomics, metagenomics and transcriptomics signatures in-stead of single omics approaches. Also, integrative genomic, epigenomic, and metabolomic analyses have better characterized post-prandial lipidomic features as potential biomarkers of nutrient intakes (including polyunsaturated fatty acids and B-complex vitamins) and subsequent cardiovascular and cancer risks. Moreover, the combined application of metagenomics, epigenomics, metabolomics, and proteomics methodologies have allowed to discriminate different dietary patterns including vegan, vegetarian, and omnivore regimes and implications in health status. In addition, the early programmation of chronic diseases in adulthood can be tackled by precision nutrition strategies targeting the epigenome, transcriptome, and the metabolome in preconception and pregnancy stages”.
Indeed, our aim was to highlight that while personalized nutrition consider unique, particular, and singular homeostatic features of each individual, precision nutrition focus on interpretation of multiple data, including omics signatures, to provide a global health/disease overview in an orchestrated manner.
Reviewer 2 Report
The manuscript „Holistic integration of omics tools for precision nutrition“ by Omar Ramos-Lopez et al. is correctly written and can be accepted after minor corrections.
1) instead of "In this cutting-edge review", it should be just "In this review"
2) The paper cannot end with a picture (Figure 2). This picture is quite complex with a lot of interrelated terms, and it needs to be better described and explained in the text; and the Conclusion part should be finished with a text.
3) As I saw, the correlations in the paper (ref 16) are not particularly significant (sentence from line 96, onwards). I suggest the author to take a closer look at that work (ref. 16) and correct the impression/description of that work.
The correlations shown in ref. 16 on Figures and the values of R2 (0.02, 0.11, 0.15, 0.26, etc) are not so high that one could talk about a prediction - my opinion.
Author Response
Reviewer 2
Comments and Suggestions for Authors
The manuscript „Holistic integration of omics tools for precision nutrition“ by Omar Ramos-Lopez et al. is correctly written and can be accepted after minor corrections.
1) Instead of "In this cutting-edge review", it should be just "In this review"
Response: The sentence “in this cutting-edge review” was now changed to “in this review” according to your suggestion.
2) The paper cannot end with a picture (Figure 2). This picture is quite complex with a lot of interrelated terms, and it needs to be better described and explained in the text; and the Conclusion part should be finished with a text.
Response: Thank you for your comment. Figure 2 was moved to the text and a more detailed description was now provided: “In this review, we provide examples of system biology studies using multi-omics methodologies (comprising genomics, metagenomics, epigenomics, transcriptomics, proteomics, and metabolomics) and interactions with dietary/nutritional factors to unravel novel insights into the mechanisms and pathways connecting the genotype to clinically relevant traits and therapy outcomes for precision nutrition applications in health and disease. In this regard, complex relationships between the genetic background, inherited epigenetics, nutrient utilization, microbiome-derived metabolites, and lifestyle factors may modulate gene and protein expressions, with implications in physiological processes, inflammation and metabolic phenotypes (Figure 2)”.
3) As I saw, the correlations in the paper (ref 16) are not particularly significant (sentence from line 96, onwards). I suggest the author to take a closer look at that work (ref. 16) and correct the impression/description of that work. The correlations shown in ref. 16 on Figures and the values of R2 (0.02, 0.11, 0.15, 0.26, etc) are not so high that one could talk about a prediction - my opinion.
Response: We agree with the reviewer. This sentence was now modified: “Although caution should be exercised, an exploratory analysis suggested that the maternal FADS2 rs174575 genotype, combined with DNA methylation status in this gene, could be related to plasma fatty acid concentrations in toddlers [16]”.
Round 2
Reviewer 1 Report
Thank you to the authors for taking into consideration my comments. I believe it improved the manuscript.